# 3D-Printed Biodigital Clay Bricks

**DOI:** 10.3390/biomimetics6040059

**Published:** 2021-10-07

**Authors:** Yomna K. Abdallah, Alberto T. Estévez

**Affiliations:** 1iBAG-UIC Barcelona, Institute for Biodigital Architecture & Genetics, Universitat Internacional de Catalunya, 08017 Barcelona, Spain; y.archimoon90@gmail.com; 2Department of Interior Design, Faculty of Applied Arts, Helwan University, Cairo 11111, Egypt

**Keywords:** clay bricks, 3D-printed bricks, 3D printed architecture, fractal dimension, form finding, biolearning, biodigital, reaction diffusion, shortest path, sustainability, material consumption

## Abstract

Construction materials and techniques have witnessed major advancements due to the application of digital tools in the design and fabrication processes, leading to a wide array of possibilities, especially in additive digital manufacturing tools and 3D printing techniques, scales, and materials. However, possibilities carry responsibilities with them and raise the question of the sustainability of 3D printing applications in the built environment in terms of material consumption and construction processes: how should one use digital design and 3D printing to achieve minimum material use, minimum production processes, and optimized application in the built environment? In this work, we propose an optimized formal design of “Biodigital Barcelona Clay Bricks” to achieve sustainability in the use of materials. These were achieved by using a bottom-up methodology of biolearning to extract the formal grammar of the bricks that is suitable for their various applications in the built environment as building units, thereby realizing the concept of formal physiology, as well as employing the concept of fractality or pixilation by using 3D printing to create the bricks as building units on an architectural scale. This enables the adoption of this method as an alternative construction procedure instead of conventional clay brick and full-scale 3D printing of architecture on a wider and more democratic scale, avoiding the high costs of 3D printing machines and lengthy processes of the one-step, 3D-printed, full-scale architecture, while also guaranteeing minimum material consumption and maximum forma–function coherency. The “Biodigital Barcelona Clay Bricks” were developed using Rhinoceros 3D and Grasshopper 3D + Plugins (Anemone and Kangaroo) and were 3D printed in clay.

## 1. Introduction

As an urgent worldwide demand, achieving sustainability in the construction sectors has become a vital need. As the construction sector is responsible for the consumption of almost half of the raw materials and energy of the planet [1,2], resulting in a great impact on the depletion of the planet nonrenewable resources, as well as the emission of greenhouse gas from the combustion of fossil fuel [2,3]. It is also undeniable that the construction industry is a major sector of the contemporary world’s economy [4], consisting of a USD 10 trillion industry and accounting for 13% of the world’s GDP in 2018; this industry is projected to increase to USD 14 trillion by 2025 [5] In the construction realm, as an important part of the world’s economy, it is wise to reduce the materials and processes that are involved in the construction practice. This concept is essential in designing construction processes, techniques and durability of the resulting structures.

In essence, minimizing material usage and construction time has provoked the emergence of pre-fabricated solutions that have been reintroduced recurrently along with different architectural trends with different available design and fabrication technologies, for example, the American Balloon structure systems [6] that are based on assembling pre-fabricated units. These pre-fabricated construction solutions emerged to achieve time-process sustainability. However, these methods belonged to the Zeitgeist of their times, solving the challenges of their times. When it comes to the contemporary construction industry, the situation does not seem the same. Reusing the same lengthy traditional construction processes, building with concrete column–beam systems that utilize incompatible materials with the environment or their time [3]. Since the arrival of clay bricks in the 1980s, their market has started to decrease, due to construction systems that are based on exterior enclosures of concrete blocks [2]. These systems are also dependent on conventional dense bricks that lack intelligence in their structural density distribution, nor do they possess any particular formal adequacy for their structural function [2,7]. This conventional brick form defies multiple spatio-physical laws by being limited to a stiff bounding box that cannot generate an optimum minimal surface, neither on its own nor in its replicative group. This is evidenced by the technological barriers caused by the inability to use these conventional bricks in low height buildings due to weight limits, as well as their environmental inadequacy as they fail to achieve thermal insulation, requiring further processes and materials to achieve insulation [8]. As well as the formal limitation, limiting the architectural forms to specific orthogonal forms that can be achieved with these primitive orthogonal conventional bricks.

On the other hand, the rapid advancement in digital design and fabrication techniques has been able to break these rigid, limited forms by adopting various strategies of digital fabrication, especially additive manufacturing, and developing cohesion between the design initiation, simulation, and execution accurately. Furthermore, nowadays, 3D printing is gaining more attention from researchers and stakeholders in the construction industry due to simplification of the construction process into a one-step printing procedure with the ability to print with a wide array of materials that can be synthesized, tuned, and customized to fully control the 3D printing material deposition. Basically, 3D printing, is an additive manufacturing technology that is able to produce complex shape geometries from a 3D model on a layer-by-layer basis. It has the potential to reduce material waste, decrease labor cost and facilitates fast production [9,10] It is undeniable that these full-scale, 3D-printed buildings (bio-plastic micro-homes, The Netherlands, DUS Architects; Tecla, Italy, Mario Cucinella Architects and WASP; Project Milestone, The Netherlands, Eindhoven University of Technology, etc.) [11,12] are major steps in integrating 3D printing as an applicable construction tool in the built environment. However, the feasibility of their mass production, along with cost effectiveness and real functional architectural programs, remains a significant challenge due to high cost of full-scale 3D printers and robotic arms, in comparison to disk-top or limited-scale 3D printers. The average clay 3D printer’s cost is between USD3,000 and 10,000, and it is easy to purchase online and can be used multiple times with no charge except for the construction material. [13], whereas the cheapest construction via 3D printing for one building costs between USD4,000 and 9,000, which only accounts for the construction material cost, not taking into account the high machinery cost that is provided only in special price quotes to large construction companies for only one 3D-printed building (as in Austin and Tabasco Houses, Austin, Texas; Tabasco, Mexico, 2018–2020, by Icon, and Gaia, Northern Italy, 2018, by Wasp) [14]. A considerable number of these examples are stuck in the phase of being mockups, and more research is pending regarding their interaction with the user, the environment, and physical forces and conditions. The majority of these examples follow the igloo house model that lacks real operative architectural programming or the ability to expand to encompass more than the simple singular function of each of these projects. Moreover, using 3D printing of concrete for houses and villas has better prospect than construction of large structures [15], due to various deficiencies such as limited capacity of 3D printer for high-rise buildings, insufficient printing materials (especially for load-bearing components), low level of customization, and complexity involved in the information processing from design to tangible objects. [15,16]. However, 3D printing has been successfully explored in printing pedestrian bridges, quick building of disaster relief shelters, structural and non-structural elements with complex geometries, printing molds for load bearing components, etc. [15,16,17].

Thus, there is a trial to exploit the maximum developments in 3D printing as a sustainable construction tool in terms of process count and material usage. As, among the numerous prospects of 3D printing in construction that were reported in recent studies, cheaper construction, reduced material use, improved safety, less reliance on human resources, better branding and market share, and durable and sustainable construction [9,10,18,19]. In this work, we propose the pixelation or the tessellation of the architectural construction process by 3D printing building units or bricks to attain sustainability through two main aspects: minimum material usage and formal sufficiency through fractal dimensions. A bottom-up methodology justifies this proposal of the fractal dimension aspect of 3D-printed architecture as it exploits the advantages of both the modular collaborative construction system (bricks) and 3D-printed architecture techniques.

Thus, the objective of this work was to employ form intelligence in the bricks’ design to achieve sustainability (less material and various formal iterations) in a single brick and in its replica while maintaining the smallest amount of material consumption and fabrication processes as possible. This article focuses on the state-of-the-art of clay as the building material of bricks, including the chemical and physical properties of clay, full-scale vernacular clay architecture examples and their limitations, contemporary clay bricks, and examples of digitally designed clay bricks. Algorithm-aided design and 3D printing were employed in the design and fabrication of the “Biodigital Barcelona Bricks”. These bricks were designed using a reaction–diffusion algorithm simulating the reaction of water diffusion in a clay brick based on its hydrophilic nature and connecting the latest point in water absorption in this clay brick through the shortest path algorithm to achieve material density optimization that is informed by natural models. Then, this optimized brick design was manipulated through different spatial iterations following the formal physiology of each design case, as well as achieving sustainability in the construction process by using 3D printing on a fractal scale. Representing 3D-printed bricks as “building units”, “pixels”, and “collaborative construction systems”: all these concepts take into account a similar sense of the definition of bricks while depending on different criteria. “Building units” term was used as the constructional definition of bricks. “Collaborative systems” term was used in the sense of biological references of similar systems that are composed of many assembled smaller parts or agents, and “pixels” term was used in the sense of digital interpretation of a surface or a volume. All these terms explain the “fractal dimension” that was pivotal to accomplishing the objective of this paper. This 3D printed architecture pixilation offers a vanguard method to popularize, democratize, and facilitate the usage of 3D printing as an efficient and affordable tool in the architectural construction realm, avoiding the high cost of full-scale, one-step, 3D-printed architecture or uncontrolled material deposition, thereby achieving 3D-printed architecture in a flexible and sustainable way, in addition to its endless formal iterations. A diagram describing the research methodology and sections is provided in Figure 1.

## 2. Clay Bricks in Architecture

Clay is one of the most abundant and traditional building materials in the world. Given the length of time it has been used for, building with clay has always been a cost-effective, easy, and sustainable approach. The proof of this lays not only in the fact that clay is a very common substance, but also that between one-half and two-thirds of the world’s population still live or work in buildings made with clay, often baked into brick, as an essential part of their load-bearing structures [20,21].

The defining mechanical properties of clay are its plasticity when wet and its ability to harden when dried or fired. Clays show a broad range of water contents within which they are highly plastic, from a minimum water content (called the plasticity limit) where the clay is just moist enough to mold, to a maximum water content (called the liquid limit) where the molded clay is just dry enough to hold its shape [22]. High-quality clay is also tough, as measured by the amount of mechanical work required to roll a sample of clay flat. Its toughness reflects a high degree of internal cohesion [22].

Clay has a high content of clay minerals that give it its plasticity. Clay minerals are hydrous aluminum phyllosilicate minerals composed of aluminum and silicon ions bonded into tiny, thin plates by interconnecting oxygen and hydroxyl ions. These plates are tough but flexible, and in moist clay, they adhere to each other. The resulting aggregates give clay the cohesion that makes it plastic [23]. When the clay is dried, most of the water molecules are removed, and the plates’ hydrogen binds directly to each other so that the dried clay is rigid but still fragile. When the clay is fired to the earthenware stage, a dehydration reaction removes additional water from the clay, causing the clay plates to irreversibly adhere to each other via stronger covalent bonding, which strengthens the material.

The tiny size and plate form of clay particles give clay minerals a high surface area. In some clay minerals, the plates carry a negative electrical charge that is balanced by a surrounding layer of positive ions (cations), such as sodium, potassium, or calcium. If the clay is mixed with a solution containing other cations, these can swap places with the cations in the layer around the clay particles, which gives clays a high capacity for ion exchange [23]. The chemistry of clay minerals, including their capacity to retain nutrient cations such as potassium and ammonium, is important for soil fertility.

On the other hand, composing a bio-based material requires concrete knowledge of what the base should be and what the filament should be and, in between them, what can enhance their coherence and total physical, chemical, and mechanical properties. Based on the mentioned properties of clay, it is the most prominent candidate for serving as a base material due to its high surface area as well as being the most widely available mineral possessing unique crystal structures, cation exchange capabilities, plastic behavior when wet, and catalytic abilities. Moreover, clay is a bio-receptive material that boosts the growth of living organisms such as plants and algae. This bio-receptive capacity of clay bricks has even been discussed in association with conventional construction systems using bricks and mortar [24], where the high moisture content of both bricks and mortar has been described as a pivotal criterion in boosting the bio-receptivity of these materials. An example of bio-receptive clay building units is the precedent in Richard Beckett’s “Bio-receptive Tile Bricks”. Another example was communicated in [25] work, proposing biomimetic place-based design (BPD) as an approach for integrating biological strategies developed by ‘champion adapters’ within the ecosystems via translating their logics into design principles for built environment design and engineering. This approach supports a design that is locally attuned, adaptable, and resilient to local operating conditions and challenges [25,26]. These properties enable greater application in many industries and clay-based materials. Thus, building with clay has always been a sustainable, cost-effective method due to its very low environmental impact, easy excavation, and simple processing and production methods. However, building with clay according to common methods is not easy, nor is it standardized for applications on a large and rapid scale. For example, the Mousgoum building technique in Africa [27] is an example of sustainable clay buildings having a very solid construction, as they have thicker walls at the base and thinner walls at the top, which enhances the structure’s strength while being highly textured to allow individualization of the surface, offering a drainage function, natural thermal insulation, and passive cooling due to clay’s hydrophilic nature. However, these Mousgoum structures require frequent maintenance of the coating, and they are unsuitable for use in rainy climates. Another example of sustainable clay architecture is exhibited in Hassan Fathy’s earthen architecture in Egypt. Fathy’s Nubian clay domes and the unique architectural programming nourished and influenced by the Islamic architectural passive cooling elements have enabled the design and construction of a vernacular clay architecture that is environmentally friendly and sufficiently sustainable [28]. Figure 2 exhibits the vernacular clay architecture in Africa, showing simple building technologies that are mainly based on hand labor.

However, these vernacular clay architecture examples are the results of their time and the technology available for material synthesis and construction techniques. Not to mention their insufficiency in material consumption control, or the lengthy time they require for building a single housing unit. In our current biodigital age where rapid advancement is occurring equally and in parallel in both in biotechnology and digitalization, the synthesis of clay materials and adequate physiological design forms are informed by digital data. These digital tools have enabled the direct insertion and realization of biological physiological forms that are biolearned from nature [30], thereby solving the relevance of the physiological form of a material structure from the micro to the macro level through topology, mechanical simulation, and formal optimization, accordingly. Few recent design projects have tackled this relation between 3D printed Brick’s formal design and sustainability. For example, the Twisted Tower developed by the Plasma Studio, that used robotic printing arms to 3D print 2000 3D-printed terracotta bricks [31]. Another experimentation on formal design optimization to achieve structural efficiency is evident in “Building Bytes 3D printed Bricks” by Brayan peters that has adapted a desktop 3D printer to produce ceramic bricks for building architectural structures. Predicting that 3D printers will become portable, inexpensive brick factories for large-scale construction, with the implementation of several 3D printers that could work simultaneously on site using pre-made or locally manufactured material [32]. Despite these previous trials to optimize the formal design of a brick to achieve structural efficiency, they lacked the biolearning reference of formal physiology. As the majority of the reached bricks’ forms resulting from these studies, did not follow biological logics that are related to the typical design case. Furthermore, 3D printing in Clay is still in need for further experimentation, as it undergoes significant alteration before, during and after 3D printing. Material properties of clay as viscosity, color, and texture, as well as printing sittings and tools (i.e., 3D printer, extrusion nozzle, tool paths, etc.) are among the variables that have major effects on the form resolution of the final outcome. This way, 3D printing with clay still requires formal customization to attain maximum resolution with minimum material deposition [33]. Thus, in the current work and in the following section, we propose formal physiological optimization for the material deposition, based on the specific customized logic of the physical and chemical properties of clay, using a biodigital design method.

## 3. Form Physiology of the Biodigital Clay Brick (Reaction Diffusion and Shortest Path)

From a biolearning point of view, physiology is the scientific study of functions and mechanisms in a living system. Physiology focuses on how organisms, organ systems, individual organs, cells, and biomolecules carry out their chemical and physical functions in a living system [34], whereas histology is the microscopic anatomy or microanatomy of biological tissues. Looking deeply into these two concepts, a strong bond always exists to relate the shape or form to its function. Human beings have always been subconsciously aware of this interdependent form–physiology relation, and they have applied it in designing their lives, from the simplest utilities to the largest shelters. Of course, the maturity, scale, and level of processing and integrating this concept in the design process has always been dependent on the level of invasion and magnification of visualization and analysis of the natural source of inspiration. This multi-scale biolearning started by the mere imitation of successful bio examples that applied the concept of formal physiology on a “macro” scale, for example, Gaudi’s Casa Batllo Bony Facade. Moving forward in time and in technological advancement, the level of invasion and magnification of these sources of inspiration has matured the biolearning process and enabled deep understanding and integration of the concept of formal physiology, from design to production. Complying with this concept, the Biodigital Barcelona Bricks presented in this work applied formal physiology in their formal design to achieve sustainability through minimum material usage and form-function adequacy. In the current study, a digital simulation was applied to perform a two-step simulation of the reaction–diffusion behavior based on clay’s hydrophilic properties and water content diffusion through its particles, as well as tuning the structural coherency of the resulting forms by applying the shortest path algorithm to minimize the distance between every two bearing points in the brick’s design, resulting in a distributed load effect. As implied from the structural function of the bricks, the function can be broken down into its physical and mechanical requirements [35]. The first of these requirements is the load distribution avoiding stress being concentrated over specific points. The second is achieving maximum coherency and strength with the minimum amount of material. Figure 3. Exhibits the followed methods to design and fabricate the proposed biodigital bricks.

### 3.1. Form Physiology: Biodigital Form-Finding, Simulation, and Optimization

Recently, achieving sustainability in the built environment takes the track of material customization and control. This is focused on developing the formal design of clay-based materials that open both the possibilities of integrating biodigital generative forms of digital DNA that is the algorithms or digital codes that are responsible for specific formal phenotypes that are extracted from the shape grammars of the original biological reference, or via hosting natural organisms or parts of them (as bacteria, fungi, algae, cells, genes, etc.) in an autonomous way to attain generative behavioral materials. These materials also facilitate the shape-changing of architectural elements, enabling the design process to include real-time morphogenesis by integrating an organism that grows and interacts with a consortium of materials, that compos its microenvironment. This aspect opens infinite applications of self-healing, clay-based materials and morphogenetic architecture in real performance, which are still a novel multidisciplinary field that requires further contributions.

Clay bricks in our biodigital age and by biodigital means, aims to achieve formal physiological sufficiency using a biolearning method while maintaining ease of implementation, mass production, and standardization. These were applied in the design and fabrication of the Biodigital Barcelona Bricks that were generatively designed using a form-finding process utilizing branching and reaction–diffusion algorithms to simulate the physical reaction of aqueous diffusion in clay. Reaction–diffusion systems are mathematical models which correspond to several physical phenomena. The most common of these is the change in space and time of the concentration of one or more chemical substances due to local chemical reactions and diffusion, which causes the substances to spread out over a surface in space. Reaction–diffusion systems are naturally applied in chemistry, biology, geology, and physics. Physically, the reaction–diffusion model describes the emergence of periodic patterns such as spots, stripes, and mazes through chemical interaction among cells [36]. In the current work, this was applied as a first step of form finding a simulation. Through this simulation, the latest regions to absorb water (in clay) in the diffusion reaction were then connected through a branching algorithm of the shortest path, which, at the same time, describes a behavioral growth pattern found in nature. The shortest path, or Dijkstra’s algorithm, is an algorithm for finding the shortest paths between nodes in a graph, which may represent, for example, road networks. [37], and as the name of the algorithm suggests, the “Shortest Path” was developed congruently with the structural function of these bricks, shortening the span between each bearing’s solid point in order to create a distributed load effect. In structural mechanics, the distributed load is a load that is distributed continuously over a given area or along a given line. A continuous load may be uniformly distributed, having a constant intensity, or vary according to some specific patterns [38]. Thus, this shortest path algorithm applied in the current brick’s form finding is congruent with achieving the distributed load by minimizing the distance between each two points in the population of points composing the 3D space of this brick. These form-finding simulations were performed in Rhinoceros 3D [39,40] (Anemone, and Kangaroo and fabricated by 3D printing by Noumena in Barcelona from November 2020 to March 2021, with the measurements of the traditional Catalan manual ceramic brick (4.5 × 14.5 × 29.5 cm).

As an initial step, the reaction–diffusion algorithm was employed to generate a diagram of the latest points to absorb water or liquids based on the clay hydrophilic nature. Given these maps that specify where the latest points to absorb liquids are, a shortest path algorithm in Anemone was used to connect these points. Figure 4 shows the diagram of the shortest path algorithm connecting the reaction–diffusion’s latest points to absorb water.

The third step was to examine these resulting shortest paths forms for the final brick resistance to cracking by applying a standard load simulation in Kangaroo, resulting in different design iterations: V1, V2, and V3. The optimum design iterations for structural behavior were selected to be further tested for various material densities, varying the density of each brick in a bulk brick model from 25%, to 55%, 75%, and 95%, and a linear or hollow brick model starting from the thickness of 0.25 cm, 0.55 cm, 0.75 cm, and 0.95 cm for each iteration (V1, V2, and V3), as shown in Figure 5, Figure 6 and Figure 7.

This density variation in the biodigital bricks’ iterations enabled the actual study of the used material amount in these physical models, as well as their resistance to cracking. A following research paper will exhibit in detail further mechanical tests to prove the structural efficiency of the proposed forms of the biodigital bricks. This 3D printed material examination customization was also reinforced by comparing to other digital fabrication strategies that will be exhibited in the following section, to prove the competence of 3D printing to achieve sustainability in material usage and production processes.

### 3.2. Digital Fabrication: 3D-Printed Pixelated Architecture

Testing different digital fabrication strategies and materials aimed to test the potential of mass production and the sustainability of material processing on the broader scale of the construction industry. Both subtractive and additive digital fabrication methods were tested using laser cutter, and 3D printing. The tested laser cutter method was employed to generate contoured or stacked layers of the biodigital bricks from different types of board wood. These layers were cut in negative and positive profiles and contoured in different thicknesses and manipulated by varying the space between each layer and the following from zero to an equal thickness of the used wood board as shown in Figure 8. These empty spaces were designed to facilitate vertical gardening and to be able to host soil for growing different plants. These stacked, contoured digital fabrication strategies also enabled permutations in the grouping together of various biodigital bricks’ profiles, resulting in more formal sophistication as well as enabling the stacking of different profiles from the same design iteration in four different orientations (front, back, right, and left) (Figure 9).

This subtractive digital fabrication strategy proved to be easily adopted and cost-effective, as well as opening more potentials for formal manipulation as shown in Figure 6 and Figure 7 and being lighter in weight than the 3D-printed clay bricks. However, these wooden stacked bricks were not the optimal sustainable solution as they needed to be glued or joined together through a specially designed joinery system that requires more material to be processed, and consequently more fabrication processes. Thus, the additive digital fabrication method using 3D printing was tested next.

The 3D-printed bricks were printed in standard clay material in an ambient room temperature. The printing process was conducted in collaboration with Noumena as shown in Figure 8. The different 3D-printed design iterations of the biodigital bricks revealed that the linear bricks’ model in all design iterations showed enhanced resistance to cracking. These linear models in the three design iterations (V1, V2, V3) reached the minimum usage of material, 50% less material, and were lighter in weight and used less printing time than the bulk brick models in all the design iterations (Figure 10). Furthermore, the linear models also exhibited cracking resistance in comparison to the bulk models in which some of them suffered from cracking in certain points. Thus, the digital simulation of the bricks’ design topology and simulation were congruent with physical models’ experimentation, achieving an optimized formal design that achieves minimum material consumption and minimum printing time and processes.

## 4. Conclusions

The current work aimed to make a sustainability leap in construction and 3D-printed architecture. This change was achieved by using both less material and fewer processes through minimizing material consumption and material processing while maintaining highly functional formal design. This was attained by the employment of the digital advancements in algorithm-aided design, structural simulation, and 3D printing technologies. The algorithm-aided design enabled the physiological form-finding of the bricks’ forms as construction pixelization. The biodigital brick followed its specific function, resulting from the reaction–diffusion algorithm that simulated the diffusion of liquids (water) in clay materials to specify the last zones to absorb water in clay. These points were further optimized following the shortest path algorithm to minimize the span between each two points to guarantee equally distributed loads and enhanced resistance to cracking. This form-finding simulation contributed to the novelty of the proposed concept of pixelating the architectural 3D printing and is a further step in the democratization the dominant trend of celebrating the advancements in 3D printing tools in the construction realm. This avoids the drawbacks of one-step printing of full-scale architecture, which does not contribute to democratizing the practice and is the least efficient in terms of material consumption, process, and cost. The proposed pixelation of architectural 3D printing coupled with an optimized formal design and a minimal amount of material and processes provide greater potential to utilize 3D printing as a sustainable tool in the architectural construction realm. Moreover, the simulation and the 3D printing experiment proved that the linear model of the Biodigital Barcelona Bricks iterations V1, V2, and V3 achieved optimum coherence with 50% fewer materials and less printing time, which achieved the objective of this research.

Thus, these are not only the first proposed 3D-printed, biodigital baked ceramic bricks that were digitally designed and manufactured, but are also the first to be flexible with multiple spatial orientations and spatial organizations. This type of brick could not have been conceived, designed, or manufactured before the current advances in 3D printing and form customization; it is absolutely the product of our time, of our zeitgeist.

## Figures and Tables

**Figure 1 biomimetics-06-00059-f001:**
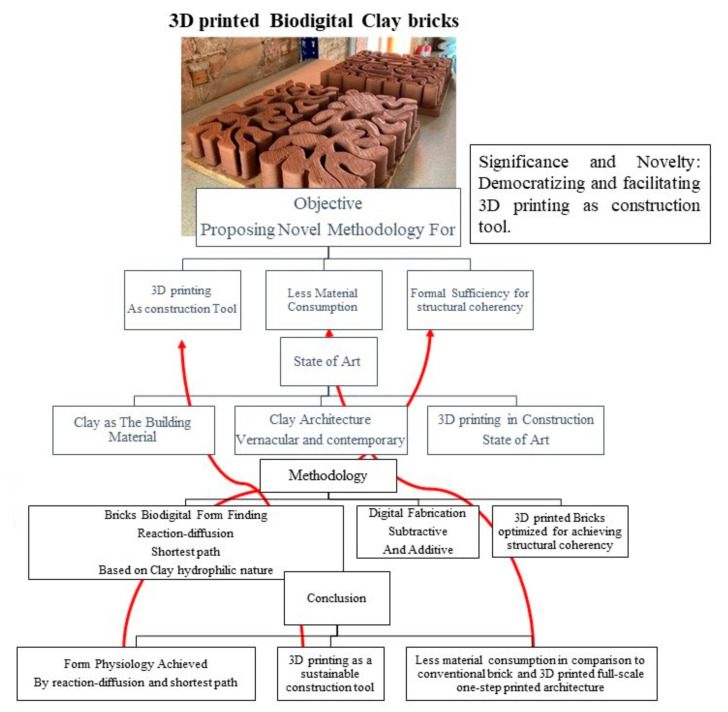
Diagram showing the objective and methodology of the 3D-printed Biodigital Barcelona Bricks. The diagram also shows the different sections of this paper’s structure and their relevance to the objective. (By the authors).

**Figure 2 biomimetics-06-00059-f002:**
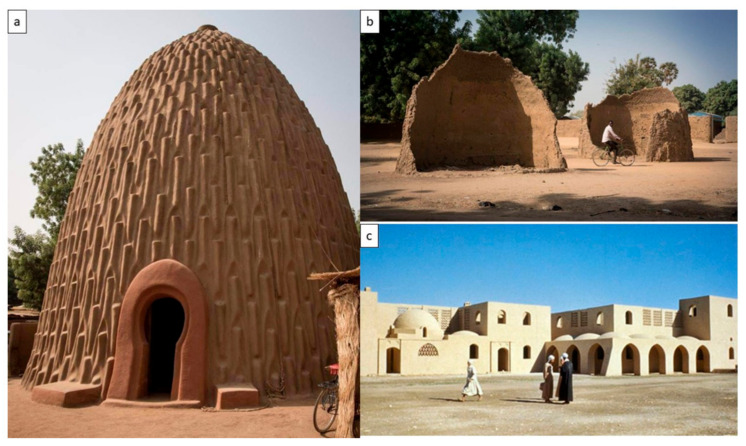
Vernacular clay architecture in Africa with simple building technologies: (**a**) shows Mousgoum houses built with clay by manual methods, while (**b**) exhibits the ruins of some of these Mousgoum houses due to cracking or poor rain resistance due to the hydrophilic nature of clay materials, and (**c**) exhibits earthen architecture by Hassan Fathy in New Gourna village in South Egypt, exploiting the passive cooling nature of clay materials through employing it in the design and construction of passive cooling Islamic architectural elements [29] (Photos: Institute of Nomadic Architecture. http://artsection.org/mousgoum.html).

**Figure 3 biomimetics-06-00059-f003:**
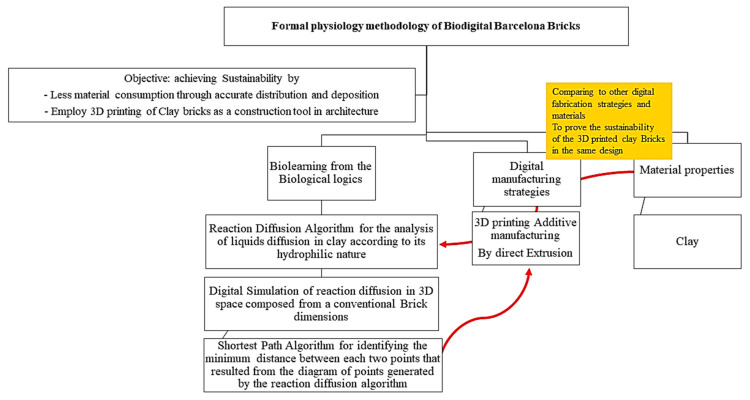
Methodology of achieving Formal Physiology of the 3D printed Biodigital Barcelona Clay Bricks.

**Figure 4 biomimetics-06-00059-f004:**
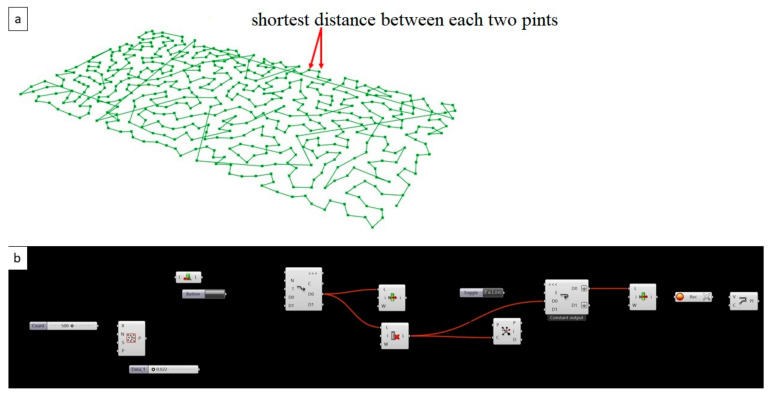
The shortest path algorithm in Grasshopper Anemone that was employed to connect the resultant latest points in the reaction–diffusion system based on the hydrophilic nature of clay. This proves the structural coherence of the bricks’ design by minimizing the span between every two points while also minimizing the material consumption due to the accurate distributed material deposition where needed. (**a**) the resultant shortest path diagram of the connected points that were the last to absorb water in the volume of a conventional Catalan brick, and were resulting from the first step simulation of reaction diffusion, (**b**) the grasshopper algorithm of the shortest path logic that was applied to connect the result points from the first step simulation, in order to find the shortest distance between these points to minimize the span between them to achieve distributed load and minimum material deposition as well. Images by the author.

**Figure 5 biomimetics-06-00059-f005:**
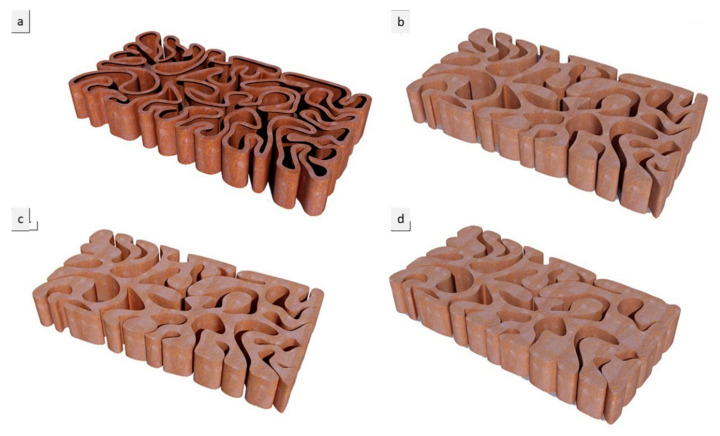
Alberto T. Estévez (Yomna K. Abdallah, computational designer), Biodigital Barcelona Bricks (3D-printed clay), GENARQ/iBAG, UIC Barcelona. V1 design iteration of the Biodigital Barcelona Bricks showing the bricks’ varied material density from a linear model of 0.5 cm thickness to a bulk model (**a**–**d**), with the densities of the brick varying from 25% to 55% and 75%, to experiment structurally different densities of the 3D-printed bricks. Images by the author.

**Figure 6 biomimetics-06-00059-f006:**
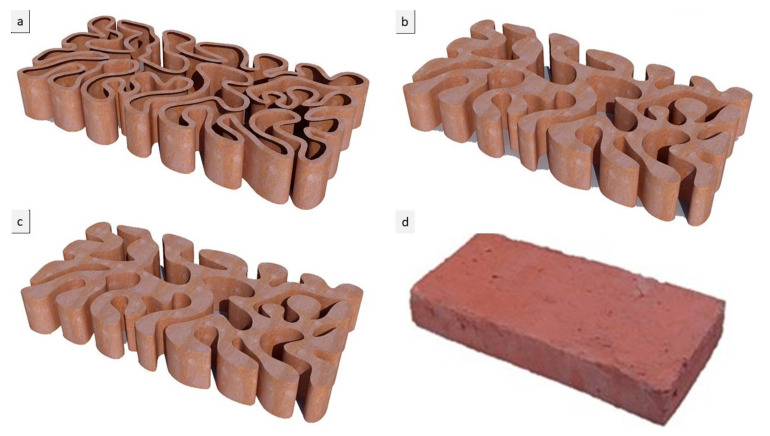
Alberto T. Estévez (Yomna K. Abdallah, computational designer), Biodigital Barcelona Bricks (3D-printed clay), GENARQ/iBAG, UIC Barcelona. V2 design iteration of the Biodigital Barcelona Bricks showing the bricks’ varied material density from a linear model of 0.5 cm thickness in (**a**) to a bulk model, with densities of the brick varying from (**b**) 25% material deposition density to (**c**) 55% material deposition density, (**d**) exhibits the original Catalan brick form and unified density in comparison to the optimized proposed Biodigital Barcelona Bricks. Images by the author.

**Figure 7 biomimetics-06-00059-f007:**
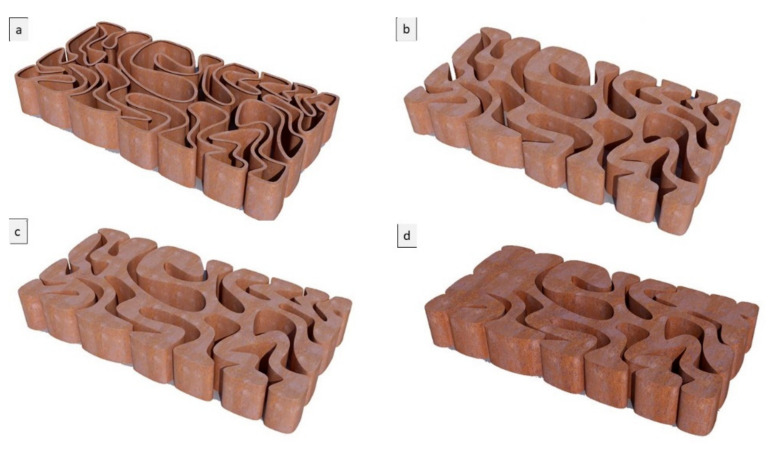
Alberto T. Estévez (Yomna K. Abdallah, computational designer), Biodigital Barcelona Bricks (3D-printed clay), GENARQ/iBAG, UIC Barcelona. V3 design iteration of the Biodigital Barcelona Bricks showing the bricks’ varied material density from a linear model of 0.5 cm thickness to a bulk model (**a**–**d**), with the density of the bricks varying from 25% to 55% and 75%. Images by the author.

**Figure 8 biomimetics-06-00059-f008:**
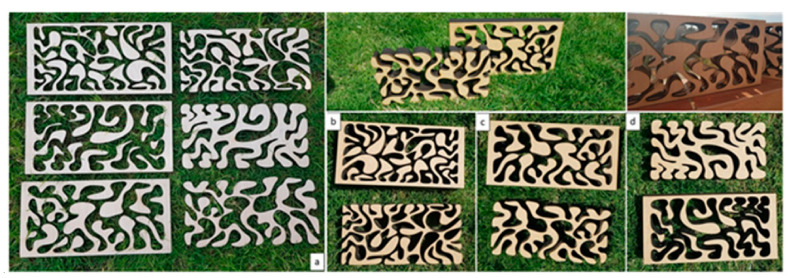
Alberto T. Estévez (Yomna K. Abdallah, computational designer), Biodigital Barcelona Bricks, GENARQ/iBAG, UIC Barcelona. The three design iterations of the Biodigital Barcelona Bricks digitally fabricated in wood using a laser cutter. (**a**) The three different design iterations digitally cut in wood by a laser cutter, resulting in more design options and providing positive and negative profiles of the brick models per each design iteration. (**b**) Design iteration V1 was fabricated following the contouring strategy of the cut profiles of the brick in positive and negative profiles. (**c**,**d**) Design iteration V2 and V3 were fabricated by contouring the model in positive and negative profiles. Photos by the authors.

**Figure 9 biomimetics-06-00059-f009:**
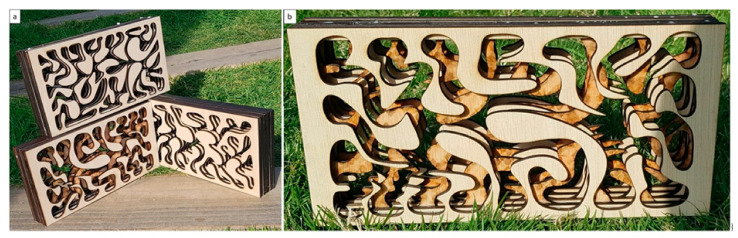
Alberto T. Estévez (Yomna K. Abdallah, computational designer), Biodigital Barcelona Bricks (laser-cut wood), GENARQ/iBAG, UIC Barcelona. Various design possibilities that emerged from varying the stacking or contouring orientation of each biodigital brick design iteration.(**a**) The achievement of these possibilities by varying the distance between each profile and then allowing the integration of soil or other biomaterials so as to be able to grow plants for vertical farms or gardens by varying the orientation of each profile (from the front, back, right, and left) and by applying permutations through combining the profiles from the three different design iterations together in one contoured brick design, as shown in (**b**). Photos by the authors.

**Figure 10 biomimetics-06-00059-f010:**
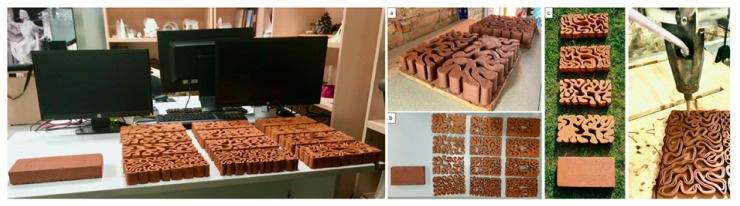
Alberto T. Estévez (Yomna K. Abdallah, computational designer), Biodigital Barcelona Bricks (3D-printed clay), GENARQ/iBAG, UIC Barcelona. (**a**–**c**) show different iterations (V1, V2, and V3) printed with clay. Right: 3D printing process of the linear model. Photos by the author.

## Data Availability

Not applicable.

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
