# Peer review of "3D-Printed Biodigital Clay Bricks"

_biomimetics, 2021, doi:10.3390/biomimetics6040059_

Round 1

Reviewer 1 Report

Dear author, thank you for your contribution. Please add relevant references in the first part of your article. Your work would benefit from a clear statement on what you are trying to do. What is the ultimate goal? If talking about bricks, I suggest considering structural issues. You mention that the voids can be used for lighting; I suggest to keen this for another paper, it is misleading and distracts from the subject. An illustration of the GH code is not necessary. Please reference scientifically and avoid copying and pasting links. Notions on democratizing design through 3D-printing are a different topic than that you are working on. If talking about bio-activity, then explain this. Explain all your figures. Be precise ad focused. Generally the article covers too many topics with too many big words. Please explain what you mean by broader scale of the construction industry. Your claim that it is the first bio-digital brick etc. is questionable. Please reference.

Author Response

The authors take this chance to thank the reviewers for their constructive comments:

ALL MODIFICATIONS WERE HIGHLIGHTED IN GREEN IN THE MANUSCRIPT.

Comment No.1: "Please add relevant references in the first part of your article."

Response No.1: Done; First this paragraph was extended as follows "The construction industry is a major sector of the contemporary world’s economy (Alaloul, et al., 2021), as it resembled 10 trillion dollars, and 13% of the world GDP, in 2018; and is projected to be 14 trillion by 2025 (McKinsey Global Institute, Reinventing Construction: A Route to Higher Productivity, February 2017), and references were added (highlighted in green), cited in the text and mentioned in the reference list.

Comment No.2:  "Your work would benefit from a clear statement on what you are trying to do. What is the ultimate goal? If talking about bricks, I suggest considering structural issues".

Response No.2: Done, the objective of the presented work was added in clear words in the introduction section.

Comment No.3: "You mention that the voids can be used for lighting; I suggest to keep this for another paper, it is misleading and distracts from the subject".

Response No.3: Done, the lighting potential of the bricks with LED light was removed from the paragraph.

Comment No.4: "An illustration of the GH code is not necessary".

Response No.4: Figure.2 was intended to show that GH code was used to find the shortest distance between two points in the resultant reaction-diffusion form, to stress on the structural efficiency of the bricks' design showing the minimum span between two points.

Comment No. 5: "Please reference scientifically and avoid copying and pasting links".

Response No.5: Done, all references were revised and modified scientifically in the reference list and in the text.

Comment No.6: "Notions on democratizing design through 3D printing are a different topic than that you are working on".

Response No.6: " This 3D printed architecture pixilation is offering a vanguard method to popularize, democratize and facilitate the usage of 3D printing as an efficient and affordable tool in the architectural construction realm.", the authors' objective is strongly related to the construction industry as part of the "World Economy", that includes both developed and developing zones that may not be able to afford the 3D printed architecture full scale, thus democratizing pixelated (units) of 3D printing as a construction tool is highly connected and is the main result of this work. Proposing the 3D printed biodigital bricks as an alternative construction solution that can be applied in these economically limited communities.

Comment No.7: "If talking about bio-activity, then explain this".

Response No.7: the word "Bioactive" was replaced by the word "Organism", explaining what is "Bioactive" will extend beyond the scope of this research. The authors only mentioned it as an option of two options to achieve formal physiology, which means the formal adequacy to achieve its designated function. However, and complying with the reviewer comments about simplifying and focusing on the paper's main objective we use a simpler word which is "organism".

Comment No. 8: "Explain all your figures".

Response No.8: All Figure captions were revised and extended in the explanation where applicable.  

Comment No. 9: "Be precise ad focused".

Response No.9: Done, the manuscript was revised, rephrased, and grammatically checked by a native English Checker.  

Comment No. 10: "Generally, the article covers too many topics with too many big words.

Please explain what you mean by the broader scale of the construction industry".

Response No. 10: Done, the full manuscript was rephrased, and reduced to specific topics that address directly a precise objective.

Comment No.11. "Your claim that it is the first bio-digital brick etc. is questionable. Please reference".

Response No.11: The authors didn't find relevant previous references that exhibit Biodigital 3D printed Bricks with the design methodology and objective of this presented work, this is why we claim it is the first to be designed and 3D printed and proposed for construction in this essence.  

Reviewer 2 Report

Authors are advised to 

  1. increase the literature review dramatically including papers from this journal and fabrication related journals to justify the significance of this research
  2. have a native speaker to edit the content thoughtfully to meet the standard of this journal.
  3. using sub-heading to structurize the paper to make the work more readable.
  4. rhino and grasshopper/with plugins are the platform that should be properly cited.

Author Response

The authors take this chance to thank the reviewers for their constructive comments 

Comment No.1: "increase the literature review dramatically including papers from this journal and fabrication related journals to justify the significance of this research".

Response No.1: Done, the state of art was expanded and supported with references highlighted in green in the manuscript. However, the authors want to emphasize that this isn't a review paper. This paper is reporting a novel methodology in biodigital design and fabrication by 3D printing to be adopted in the construction realm, which implies the direct presentation of the design methodology and results while exhibiting the relevant previous state of art briefly where needed. Lastly, the significance of this research was justified through the objective that was rephrased in the introduction section, (highlighted in green).

Comment No. 2: "have a native speaker edit the content thoughtfully to meet the standard of this journal."

Response No. 2: Done, the manuscript was revised and corrected by a native English speaker.

Comment No. 3: "using sub-heading to structurize the paper to make the work more readable".

Response No.3: Done, highlighted in green in the manuscript.

Comment No. 4: "rhino and grasshopper/with plugins are the platform that should be properly cited."

Response No.4: Done, references were added in the manuscript, and in the reference list.v

Reviewer 3 Report

From my point of view, the authors have carried out research about a really interesting issue. However, in my opinion, this text should address the following issues before being considered for publication:

  • Firstly, in my opinion, this text is an “article” instead of a “review”.
  • Secondly, the title should improve so to become easier to understand while avoid using “(“ and “/”; besides the file does not include the abstract.
  • Thirdly, in general, this text has several confusing and ambiguous parts. Its information needs to be presented, structured and justified in a better way. For example:
    • Introduction needs to be reorganized and complemented starting from general to particular (from a general overview to the specific topic), introducing this project general context and specific area of expertise, this article main contribution and novelty compared to previous scientific contributions, all related to the necessity of doing this new brick and research project. The authors should also avoid giving strong statements without evidence or references, which is the case of many sentences. Moreover, this first section lacks references in all paragraphs. Furthermore, in the last paragraph the article structure should be presented, and the authors should clearly and explicitly relate this structure to the different sections of the manuscript.
    • This introduction and all this document sections present a lot of different concepts and possibilities. Related to this, the second and third paragraphs of page 2 give several confusing meanings to the term “bricks”. The authors should clarify what this research project considers as bricks and present the related definition/s. From my point of view, the broad variety of information in this document does not improve the outcomes. Following the authors comment to “Less is more”, in my opinion, parts such as the subtractive digital fabrication strategy should be discarded for this present document. Instead, the authors should lengthier and better clarify the scope and boundaries of their research project. For example, what are these new bricks for? For load bearing walls? For cladding?
    • In this sense, the reference clay buildings in section 2 should be more related to these boundaries and case study. Brick based vernacular, industrialized and innovative references should be added.
    • As part of this improved structure, a clear framework and methodology should be given and justified, with its main steps - state of the art, design, optimization, prototyping… - and tools and methodologies - Grasshopper Anemone, Kangaroo, 3D printer. The use of these methodologies should be justified compared to other alternative methods and tools available.
    • Then, all these steps and methods should be explained in depth. For example, the shortest path algorithm should be further depicted. Also, this article highlights the contribution of this research in terms of optimizing material and structure. If so, this structural optimization should be further explained. From its definition, steps, tools… to its results. Showing evidence material. And in a way that reviewers can review it and other researchers could be able to replicate this methodology in its research. So, the authors should further explain the 2 requirements presented at the end of the first paragraph in Section 3, as well as how this research project achieves these requirements. For example: the first paragraph in page 5 should further explain the structural efficiency (which construction elements, connections between bricks, load case, stresses); and page 5 last paragraph should further explain how the algorithm works and its precise relation to the hydrophilic nature of clay.
    • Also, all properties should be better explained and properly demonstrated. For instance, “zero% waste” (pg 2, end paragraph 2) or “easily adopted and cost effective” (page 8, line 2).
    • These lacking sections and explanations are also needed so that conclusions can rely on them and on a discussions section. At present, numerous conclusions do not properly rely on this paper contents.

Other comments:

  • Some websites are neither well cited nor presented as references.
  • General English language revision: i.e. use of “the”, “processes sustainability” and “the material consumption and the material processing” (conclusions, 3rd line).

Author Response

The authors take this chance to thank the reviewers for their constructive comments 

General Note: due to the major reforming, rephrasing, tunning, and extension on specific points. All the corrections are highlighted in green in the manuscript.

Comment No.1: Firstly, in my opinion, this text is an “article” instead of a “review”.

Response No.1: yes, the manuscript is reporting a novel digital design methodology and fabrication through 3D printing to be adopted in the construction realm. It is not a review.

Comment No.2 " Secondly, the title should improve so as to become easier to understand while avoid using “(“and “/”; besides the file does not include the abstract."

Response No.2: Done, the Title was changed to " 3D printed Biodigital Barcelona Bricks (Material Sustainability and Structural Efficiency).

The abstract was added and modified to convoy with the following structure (Current state of 3D printing as a construction tool in architecture, the question of efficiency of this practice in terms of material consumption and structural efficiency, followed by the objective of the paper of proposing biodigital bricks that employ biodigital design to achieve an optimized form that achieve minimum material use and increased structural efficiency, exhibiting the methodology, digital design and fabrication tools).  

Comment No.3: "Introduction needs to be reorganized and complemented starting from general to particular (from a general overview to the specific topic), introducing this project general context and specific area of expertise, this article main contribution and novelty compared to previous scientific contributions, all related to the necessity of doing this new brick and research project. "

Response No.3: Done, the introduction was fully reorganized following the structure provided in the reviewer comments, moving from the general to the specific (starting by construction industry and economy, prefabricated systems contribution to reduce the construction time, bricks conventional construction system, new building techniques, 3D printed architecture, and lastly the explanation, justification, and novelty of the proposed project of Biodigital Bricks"

Comment No.4: " the authors should also avoid giving strong statements without evidence or references, which is the case of many sentences."

Response No.4: Done, the manuscript was all revised, and rephrased using simpler language, while omitting unevidenced sentences.

Comment No.5: " Moreover, this first section lacks references in all paragraphs."

Response No.5: Done, references were added in the text highlighted in green and added in the reference list as well.

Comment No. 6: "Furthermore, in the last paragraph the article structure should be presented, and the authors should clearly and explicitly relate this structure to the different sections of the manuscript."

Response No.6: Done, highlighted in green in the manuscript.

Comment No.7: "This introduction and all this document sections present a lot of different concepts and possibilities. Related to this, the second and third paragraphs of page 2 give several confusing meanings to the term “bricks”. The authors should clarify what this research project considers as bricks and present the related definition/s. "

Response No.7: the different concepts were reduced to the minimum only addressing the essential related aspects to this work objective and methodology. (look at response No.3, and Response No.8).

This work represents bricks as "building units", "pixels", "collaborative construction system" all these concepts are resembling bricks in the same essence but according to different criteria. "building unites" in the essence of construction definition of bricks, "collaborative system" in the essence of the biological reference of similar systems that are composed of many assembled smaller parts, and "pixels" is in the essence of digital interpretation of a surface or a volume. All these words are explaining the "fractal dimension" that is pivotal to achieve the objective of this paper.  This paragraph was added to the manuscript in the introduction. Methodology paragraph.

Comment No.8: "From my point of view, the broad variety of information in this document does not improve the outcomes. Following the authors comment to “Less is more”, in my opinion, parts such as the subtractive digital fabrication strategy should be discarded for this present document. Instead, the authors should lengthier and better clarify the scope and boundaries of their research project. For example, what are these new bricks for? For load bearing walls? For cladding?"

Response No.8: the broad variety of information was reduced to address the main objective of the paper and the project: using 3D printed bricks as a construction tool to minimized production processes, cost, and material usage. In this case the "Less is more" concept is justified, as it emphasizes on exploiting digital design and fabrication advances and 3D printing to minimize the material and deposit it where exactly needed to achieve stronger structures with less materials.

The scope of application of the bricks is still under detailed structural experimentation and will be exhibited in the following paper focused only on this aspect. The current paper is proposing the concept, the idea of using 3D printed bricks as an alternative to conventional bricks and full-scale 3D printed architecture, benefiting from the advantages of both systems without bearing their drawbacks, as high cost, uncontrolled form, uncontrolled material deposition.

Comment No. 9: "In this sense, the reference clay buildings in section 2 should be more related to these boundaries and case study. Brick-based vernacular, industrialized and innovative references should be added."

Response No.9: Done, this section was extended and supported with examples from the current state of art of advanced biobased clay bricks. Also, references were added to support the extended discussion of clay properties as a sustainable, available, and cost-effective material. Complying with the objective of this paper. (look at response No.3, and Response No.8).

Comment No.10: " As part of this improved structure, a clear framework and methodology should be given and justified, with its main steps - state of the art, design, optimization, prototyping… - and tools and methodologies - Grasshopper Anemone, Kangaroo, 3D printer. The use of these methodologies should be justified compared to other alternative methods and tools available."

Response No.10: Done, the full manuscript was extended in the description of these steps and concepts. (highlighted in green).

Comment No.11: Then, all these steps and methods should be explained in depth. For example, the shortest path algorithm should be further depicted. "

Response No.11: Done, the reaction-diffusion concept and the shortest path algorithm were further explained. (highlighted in green).

Comments No. 12: " Also, this article highlights the contribution of this research in terms of optimizing material and structure. If so, this structural optimization should be further explained. From its definition, steps, tools… to its results. Showing evidence material. And in a way that reviewers can review it and other researchers could be able to replicate this methodology in its research. So, the authors should further explain the 2 requirements presented at the end of the first paragraph in Section 3, as well as how this research project achieves these requirements.

 Response No.12: complying with previous comments of the viewer on limiting and tightening the objectives and reforming the methodology of this paper, the authors have reformed the methodology and objectives focusing on "using 3D printed clay bricks in construction by achieving formal adequacy to structural efficiency and less material usage". From a conceptual point of view that is focused on the relation between the digital design and fabrication tools using 3D printing to print these bricks. Thus, the detailed structural experimentation and results will be published in the following paper focused only on those aspects. Complying with the reviewer's comments about focusing on the methodology and objectives and avoiding the integration of multiple topics at once. (look Response No.2, No.8, and No.13).

Comment No.13: "For example, the first paragraph on page 5 should further explain the structural efficiency (which construction elements, connections between bricks, load case, stresses)".

Response No.13: The extended structural properties study of these aspects (which construction elements, connections between bricks, load case, stresses), will be included in another following paper, focusing only on the structural tests and results of these aspects. (Look response No.8, and No.12).

Comment No.14: "and page 5 last paragraph should further explain how the algorithm works and its precise relation to the hydrophilic nature of clay."

Response No.14: Done, the reaction-diffusion and the shortest path algorithm were further explained in relevance to the hydrophilic nature of clay and moist content diffusion in clay. (highlighted in green).

Comment No.15: "Also, all properties should be better explained and properly demonstrated. For instance, “zero% waste” (pg 2, end paragraph 2) or “easily adopted and cost-effective” (page 8, line 2)."

Response No.15: Done, these sentences were either omitted or justified and supported by references. Highlighted in green in the manuscript.

Comment No.16: "Some websites are neither well-cited nor presented as references."

Response No.16: Done, all websites were updated and cited properly.

Comment No. 17: "General English language revision: i.e. use of “the”, “processes sustainability” and “the material consumption and the material processing” (conclusions, 3rd line)."

Response No.17: Manuscript was revised, corrected, and rephrased.

Round 2

Reviewer 1 Report

Dear author, please clarify the relevance of this "This is focused on developing [...] or ---> hosting natural organisms in an autonomous way to attain generative behavioral materials.

Author Response

Comment No.1: Does the introduction provide sufficient background and include all relevant references?

Response No.1: Done, the introduction was further expanded and supported with references. highlighted in green in the text. 

Comment No.2: Is the research design appropriate?

Response No.2: Done, a clear structure of this research was provided in Figure.1

Comment No.3: Are the methods adequately described?

Response No.3: Done, Look at the last paragraph in the introduction. 

Comment No.4: Are the results clearly presented?

Response No.4: Done, Look conclusion section, corrections highlighted in green in the text. 

Comment No.5: Are the conclusions supported by the results?

Response No.5: Done, Look conclusion section, corrections highlighted in green in the text. 

Comment No.6: Dear author, please clarify the relevance of this "This is focused on developing [...] or ---> hosting natural organisms in an autonomous way to attain generative behavioral materials.

Response No.6: this phrase shows one specific potential of the proposed 3D digital clay bricks. showing that clay's natural properties enable it from hosting (to integrate inside it) living organisms like bacteria or fungus. when these organisms grow they will change in their form, structure, and density and they will further interact with the clay base of the material resulting in new for new physical, chemical, and structural properties of this material. however, since the authors received multiple comments on limiting the scope of the presented work and be focused on the main objective, the provided explanation in this response wasn't included in the manuscript. 

Reviewer 2 Report

Dear Authors,

thanks for submitting this article to our journal. 

The topic is relevant and the language has been improved dramatically.

However, the presentation still needs a lot of improvement in order to meet a high quality journal.

Introduction should clearly indicate the research problem that this project is going to resolve rather than mentioning the design concept of Mies and Corbusier that is not directly relevant to the solution which is a 3D print brick.

Why sustainability needs a 3D print brick with biometric design? this is still not a clear picture. Remember this is a scientific journal which needs evidence, methods or experiment and data to reify the exploration. Not just a design concept.

Pictures are evidences or enforcement for the argument or elaboration of text. Not the other way around. The captions in Pictures show a lot of information that should be extended in the main text. 

Related researches should be included in the reviews and describes the significant of this research with comparison. References should include several recent 3d print brick or clay researches such as 

  1. Shah, K.W.; Huseien, G.F. Biomimetic Self-Healing Cementitious Construction Materials for Smart Buildings. Biomimetics 2020, 5, 47. https://doi.org/10.3390/biomimetics5040047
  2. Hayes, S.; Toner, J.; Desha, C.; Gibbs, M. Enabling Biomimetic Place-Based Design at Scale. Biomimetics 2020, 5, 21. https://doi.org/10.3390/biomimetics5020021
  3. http://papers.cumincad.org/cgi-bin/works/paper/ecaade2018_104

also please reduce the web-based references if possible, otherwise please follow recent online information fashion to provide as much as information in the text as possible since online information is not a stable sources for further researchers.

Methods should be described clearly in steps or digram to allow further exploration of our readers. Also, experiments should provide the relevant dataset and parameters for rigid assessment.

Author Response

Comment No.1: Introduction should clearly indicate the research problem that this project is going to resolve rather than mentioning the design concept of Mies and Corbusier that is not directly relevant to the solution which is a 3D print brick.

Answer No. 1: The "less is more" concept was deleted. The insufficiency of the conventional clay bricks (environmental and constructional disadvantages) was extended in the introduction and supported with references. (in red in the text)

Comment No.2: Why sustainability needs a 3D print brick with biometric design? this is still not a clear picture. Remember this is a scientific journal which needs evidence, methods or experiment and data to reify the exploration. Not just a design concept.

Answer No. 2: This was expanded in the introduction with references. (in red in the manuscript).

Comment No.3: Pictures are evidence or enforcement for the argument or elaboration of text. Not the other way around. The captions in Pictures show a lot of information that should be extended in the main text

Answer No.3: all Figures' captions were revised to make sure that all the information in the captions is represented in the text as well.

Comment No.4: Related researches should be included in the reviews and describes the significance of this research with comparison. References should include several recent 3d print brick or clay researches such as 

  1. Shah, K.W.; Huseien, G.F. Biomimetic Self-Healing Cementitious Construction Materials for Smart Buildings. Biomimetics 2020, 5, 47. https://doi.org/10.3390/biomimetics5040047
  2. Hayes, S.; Toner, J.; Desha, C.; Gibbs, M. Enabling Biomimetic Place-Based Design at Scale. Biomimetics 2020, 5, 21. https://doi.org/10.3390/biomimetics5020021
  3. http://papers.cumincad.org/cgi-bin/works/paper/ecaade2018_104

Answer No.3:

Answer No.3: Done, the proposed references and other similar references (13 new references) were added to the manuscript, expanding comparisons on current projects and examples of 3D printed bricks. (in red color in the text).

Comment No. 4: also, please reduce the web-based references if possible, otherwise please follow the recent online information fashion to provide as much as information in the text as possible since online information is not a stable source for further researchers.

Answer No.4: Done, more references of papers from journals were added  (in red in the manuscript).

Comment No.5: Methods should be described clearly in steps or diagram to allow further exploration of our readers. Also, experiments should provide the relevant dataset and parameters for rigid assessment.

Answer No.5:  Done, Figure.3

Reviewer 3 Report

In the attached file

Author Response

response to reviewer comments are in the attached file 

Round 3

Reviewer 3 Report

In the attached file

Author Response

Firstly, the abstract needs improvement. Similarly to the whole manuscript, it should further focus on this specific article topic (initial design of 3d printed bricks), better explain the limitations and boundaries of this research paper, and avoid that readers get wrong expectations.

Answer: Done: (in red color in the text). 

  • - Secondly, in general, this text has several confusing and ambiguous parts. Its information needs to be presented, structured and justified in a better way. For example:
    o The introduction title is confusing.
    o There are still many strong statements without evidence or references. For example: first sentence
    of the introduction section, lines 45 to 57 from page 2, etc.
    o From my point of view, the broad variety of information in this document does not improve the outcomes. Following the authors comment to “Less is more”, in my opinion, parts such as the subtractive digital fabrication strategy should be discarded for this present document. Instead, the authors should lengthier and better clarify the scope and boundaries of their research project. For example, what are these new bricks for? For load-bearing walls? For cladding?

Done: the several topics were reduced to only relevant justified aspects of the paper topic, the title of the introduction was deleted, 13 new references were added in support of the communicated statements in the manuscript, avoiding strong expressions, subtractive digital fabrication methods were justified as a comparison to the 3D printing additive method to prove its sustainability, the application of the bricks was identified to load-bearing. 

  • In page 2, lines 73-76, the authors point out the early stage and limitations of most research articles in this area of expertise. From my point of view, the authors should also clearly present this early stage and limitations in reference to their article in all the parts of the text, from the abstract and introduction to the conclusions. This version of the manuscript is giving the wrong impression that this research project and article go beyond other former research projects.

Done: corrections were done in the manuscript (in red color).

  • Section two title should avoid undefined abbreviations. In this sense, the reference clay buildings in section 2 should be more related to these boundaries and case studies. Brick-based vernacular, industrialized, and innovative references should be added and this section should completely focus on brick-based clay architecture.

Done: the title was changed to focus on clay bricks and the content included further examples of clay bricks in architecture. 

  •  Then, all these steps and methods should be explained in depth. For example, the shortest path algorithm should be further depicted.

look section "3.1 Form physiology: biodigital form-finding, simulation, and optimization", and Figure.4.

Also, this article highlights the contribution of this research in terms of optimizing material and structure. Now that the authors have further explained it I understand that this work has focused on the bricks’ “structural optimization”/”structural efficiency”/structural coherence exclusively from the point of view of uniformly distributing the stresses during the chemical processes of its clay production. This should be clearly explained all through the article, from the abstract to the conclusions. And these explanations should clarify that the structural behavior of these bricks in any application - cladding, load-bearing walls… - has not been studied in this research article. In this sense, if a future article deals with a broader study of these brick structural performances, this structural performance cannot be commented on in this article but in the future paper.

the word "structural efficiency, coherence, etc." was removed from all the manuscript. limitations and scope of the presented work were communicated through all the different sections of the manuscript.

  • These lacking sections and explanations are also needed so that conclusions can rely on them and on a discussions section. At present, numerous conclusions do not properly rely on this paper
    contents.

Done. corrections in red in the manuscript.